

# Taxonomic, functional and expression analysis of viral communities associated with marine sponges

Mary Nguyen[1], Bernd Wemheuer[1], Patrick W. Laffy[2],
Nicole S. Webster[2,3] and Torsten Thomas[1]

[1] Centre for Marine Science and Innovation & School of Biological & Earth and Environmental Sciences, University of New South Wales, Sydney, NSW, Australia
[2] Australian Institute of Marine Science, Townsville, QLD, Australia
[3] Australian Centre for Ecogenomics, University of Queensland, Brisbane, QLD, Australia

## ABSTRACT

Viruses play an essential role in shaping the structure and function of ecological communities. Marine sponges have the capacity to filter large volumes of 'virus-laden' seawater through their bodies and host dense communities of microbial symbionts, which are likely accessible to viral infection. However, despite the potential of sponges and their symbionts to act as viral reservoirs, little is known about the sponge-associated virome. Here we address this knowledge gap by analysing metagenomic and (meta-) transcriptomic datasets from several sponge species to determine what viruses are present and elucidate their predicted and expressed functionality. Sponges were found to carry diverse, abundant and active bacteriophages as well as eukaryotic viruses belonging to the Megavirales and *Phycodnaviridae*. These viruses contain and express auxiliary metabolic genes (AMGs) for photosynthesis and vitamin synthesis as well as for the production of antimicrobials and the defence against toxins. These viral AMGs can therefore contribute to the metabolic capacities of their hosts and also potentially enhance the survival of infected cells. This suggest that viruses may play a key role in regulating the abundance and activities of members of the sponge holobiont.

## INTRODUCTION

Viruses, and in particular bacteriophages, are thought to be the most abundant biological entities in the ocean. With an estimated $10^7$–$10^8$ virus-like particles (VLPs) per millilitre of seawater, they outnumber their bacterial hosts up to 100-fold (*Breitbart et al., 2018*; *Fuhrman, 1999*; *Knowles et al., 2016*; *Parikka et al., 2017*; *Weinbauer, 2004*; *Wigington et al., 2016*; *Wommack & Colwell, 2000*). In the marine environment, viruses are considered to play an important role in: (1) global biogeochemical cycles through lysis of microorganisms (*Middelboe & Lyck, 2002*; *Weinbauer et al., 2011*), (2) microbial community structure and diversity through the 'kill the winner' process (*Angly et al., 2006*; *Winter et al., 2010*) and (3) the evolution of microorganisms through viral-mediated

Corresponding author
Torsten Thomas,
t.thomas@unsw.edu.au

genetic exchange (*Breitbart, 2012*; *Fuhrman & Schwalbach, 2003*; *Riemann & Middelboe, 2002*; *Rodriguez-Brito et al., 2010*; *Thurber et al., 2017*).

The influence of viruses on microbial community structure and evolution are of particular interest for filter-feeding reef invertebrates, such as sponges (phylum Porifera), due to their capacity to move large volumes of 'virus-laden' seawater through their bodies and their hosting of dense communities of microbial symbionts, which are likely accessible to viral infection (*Reiswig, 1971*; *Thomas et al., 2010*). Despite the potential of sponges and their symbionts to act as viral reservoirs, the investigation of sponge-associated viruses is still in its infancy. Early electron microscopy studies by *Vacelet & Gallissian (1978)* detected VLPs within parts of the sponge *Verongia cavernicola* that were devoid of choanocytes chambers, while *Claverie et al. (2009)* later predicted that the phagocytes of *Petrobiona massiliana* may be undergoing infection by a giant virus, possibly related to the *Mimiviridae*. More recently, *Pascelli et al. (2018)* detected over 50 VLP morphotypes within the cells, the extracellular matrix (mesohyl) and the associated microorganisms of 15 different sponge species.

Metagenomic sequencing has also recently been applied to taxonomically describe and functionally characterise the communities of free viruses associated with sponge species (*Jahn et al., 2019*; *Laffy et al., 2018*). This work found diverse viral assemblages that exhibited significant host species-specificity and revealed several auxiliary metabolic genes (AMGs) and viral pathogenesis pathways that were differentially enriched across sponge hosts (*Laffy et al., 2018*). AMGs encode proteins for metabolic functions that are thought to increase the fitness of the infected host, for example by allowing for a better utilisation of nutrients or energy sources. These viral communities in these studies were further dominated by bacteriophages (e.g. *Myoviridae*, *Podoviridae* and *Siphoviridae*), but had considerable variability in the relative abundance of viruses targeting eukaryotic cells (such as *Mimiviridae* and *Phycodnaviridae*). The role of viruses in host health and environmental stress responses has also been recently started to be explored and demonstrated shifts in viral community profiles (*Butina et al., 2019*; *Laffy et al., 2018*). Furthermore evidence for genetic exchange through sponge-associated viruses has been given by the discovery of eukaryotic-like genes encoding for ankyrin proteins in bacteriophages that may provide infected bacterial hosts with an increased capacity to avoid eukaryotic phagocytosis (*Jahn et al., 2019*). Several studies have also reported indirect evidence for the presence of viruses in sponges, specifically through the presence of antiviral defence mechanisms, such as clustered regularly interspaced palindromic repeats (CRISPR), that have been identified within genome and metagenome sequences of sponge-associated microbes (*Fan et al., 2012*; *Horn et al., 2016*; *Karimi et al., 2017*; *Thomas et al., 2010*). And finally, viral genomes have also been found within the microbial metagenome of a hydrothermal vent sponge from the Okinawa trough (*Zhou et al., 2019*).

While most metagenomic studies have focused on viral communities that were physically separated from their microbial or eukaryotic hosts, investigating the viral diversity that is directly associated with the cellular fraction may enhance our understanding of the role viruses play in the sponge holobiont. This study therefore aimed to investigate the abundance, diversity and functional gene repertoire of viruses associated

with metagenomic datasets derived from the microbial cells of six sponge species, including a comparative analysis with viruses associated with microorganisms from the surrounding seawater. In addition, no previous study has investigated what viruses and functions are actively expressed in sponges and we therefore used here a (meta-) transcriptomic approach for the species *C. concentrica*, *Scopalina* sp., and *T. anhelans* to fill this knowledge gap.

## MATERIALS AND METHODS

### Metagenomic, transcriptomic and metatransciptomic datasets

To retrieve viral sequences associated with microbial cells, metagenomic datasets from six marine sponges (*Cymbastela concentrica*, *Tedania anhelans*, *Scopalina* sp. from the coast of Sydney, Australia and *Cymbastela coralliophila*, *Rhopaloeides odorabile* and *Stylissa* sp. 445 from the Great Barrier Reef, Australia; July–September 2009) and corresponding seawater metagenomes previously published by *Fan et al. (2012)* were analysed (Fig. S1). Microbial cells in that study were obtained by homogenisation of sponge tissue followed by a series of centrifugation and filtration steps (3 μm cut-off), which removed sponge cells and likely removed free (i.e. not cell-bound) VLPs (*Castro-Mejia et al., 2015*; *Fan et al., 2012*). Seawater microorganism were collected by filtration on a 0.2 μm membrane. DNA from the microbial cells of each sample type were extracted, shotgun sequenced using Roche pyrosequencing technology and reads as well as assembled contigs were annotated (further details can be found in *Fan et al. (2012)*). Taxonomic and functional analyses showed high intra-species consistency and clear inter-species differences for the bacterial and archaeal communities of these sponge species that were also all distinct from those in seawater (*Fan et al., 2012*). Expressed viral genes were analysed in the transcriptomic and metatranscriptomic datasets from the sponges *C. concentrica*, *Scopalina* sp. and *T. anhelans* (sampled in July 2016), which were recently published by *Diez-Vives et al. (2017)* (Fig. S1). Transcriptomic and metatranscriptomic sequencing data were generated in that study from total RNA extractions of snap-frozen whole sponge tissue. Total RNA was separated into poly-adenylated (polyA) RNA and non-polyA RNA, with the latter being additionally depleted of rRNA, to represent enrichment of mRNA transcript of eukaryotes and mRNA of prokaryotes, respectively. These two RNA fraction were shotgun-sequenced with the Illumina sequencing platform followed by a *de novo* assembly (further details can be found in *Diez-Vives et al. (2017)*) and are here after referred to as 'transciptomic' and 'metatranscriptomic' data, respectively. The datasets combined therefore represent non-rRNA transcripts from microbial and sponge cells as well as all other constituents of the sponge holobiont (*Diez-Vives et al., 2017*).

### Identification of viral sequences

The VirSorter program (*Roux et al., 2015*) was used to identify viral sequences within reads and contigs of the cellular metagenomes and (meta-)transcriptomes mentioned above. VirSorter uses an in-built 'viromes' reference database, in which 826 proteins clusters and single genes were identified as 'viral hallmark genes.' Virsorter calculates the following parameters: (1) presence of viral hallmark genes (*Koonin, Senkevich & Dolja, 2006*;

*Roux et al., 2014*), (2) enrichment in virus-like genes (3) depletion in PFAM-affiliated genes, (4) enrichment in uncharacterized genes (i.e. predicted genes with no hits in either PFAM or the viral reference database), (5) enrichment in short genes (genes with a size within the 10% of the shortest genes of the genome or contig), and (6) depletion in strand switching (i.e. change of coding strand between two consecutive genes). Contigs were subsequently classified based on these parameters into three categories (i) category 1 ('most confident' viral predictions) for contigs fulfilling criteria 1 and 2; (ii) category 2 ('likely' viral predictions) for contigs fulfilling either criteria 1 or 2 and at least one of the other criteria; and (iii) category 3 ('possible' viral predictions) for contigs not fulfilling criteria 1 or 2, but at least two of the other criteria with at least one significance score greater than 4 (for more details see (*Roux et al., 2015*)). In this study, only contigs that were assigned to category 1 and 2 were used for further analyses. The viral sequence identified in the cellular metagenomes and (meta-)transcriptomes are available as the Supplemental Data file 'Supplementary_data_viral_contigs_transcripts.fna' with the ID description given in the Supplemental Information.

Viral sequences from the cellular metagenomes were normalised to read coverage as well as the estimated number of prokaryotic genomes (*Fan et al., 2012*). The number of prokaryotic genomes per sample was inferred using the average abundance of 82 single-copy genes, which were present in ≥90% of all complete archaeal and bacterial genomes in the NCBI RefSeq database (archaea = 261, bacteria = 7,315; July 2017) and were found as single-copies in ≥95% of all genomes. The PFAM and TIGRFAM HMM profiles (version 31.0 and 14.0, respectively) of the proteins encoded by these single-copy gene were used for homolog identification in the metagenomes using HMMsearch v3.1b2 (minimum score of 40 and maximum $E$-value of $10^{-5}$). Viral sequences in the (meta-)transcriptomes were normalised using transcripts per million (TPM) counts (*Diez-Vives et al., 2017*).

**Taxonomic and functional annotation**

Taxonomic classification of viral sequences was conducted using MetaVir2 (*Roux et al., 2014*) using the viral Refseq protein database from NCBI with BLASTp and an E-value threshold of $10^{-3}$ as well as the PFAM database v27 with HMMScan and a score threshold of 30. Taxonomic classification was based on the lowest common ancestor (LCA) affiliation of each contig as implemented in MetaVir2 (*Roux et al., 2014*). The LCA affiliation considers multiple hits on a single sequence (up to five, if available) and affiliation is made at the highest common taxonomic level from the selected genes against the RefSeqVirus database.

For functional annotation of sequences, ORFs were predicted using Prodigal v.2.6.2 (*Hyatt et al., 2010*). Predicted protein sequences were functionally classified using Diamond searches (*Buchfink, Xie & Huson, 2015*) against the COG (2014 release) (*Galperin et al., 2014*) and KEGG (October 2017 release) databases (*Kanehisa et al., 2002*) with an $E$ value of $<10^{-10}$. In addition, predicted proteins were compared to the PFAM v31 (*Finn et al., 2015*) and TIGRFAM v15 (*Haft, Selengut & White, 2003*) databases using HMMsearch with an $E$-value of $<10^{-10}$. Functional profiles for each sample are represented

**Table 1 Summary of the number of viral sequences found in each sample metagenome. A, B and C indicate sample replicates for sponges and 01, 02 and 08 indicate seawater replicates (labelling is based on *Fan et al., 2012*).**

| Sample | No. of total sequences | No. of viral sequences | % Of viral sequences in metagenome | Average length of viral sequences (bp) |
|---|---|---|---|---|
| Seawater 01 | 18,797 | 88 | 0.468 | 644 |
| Seawater 02 | 24,581 | 109 | 0.443 | 957 |
| Seawater 08 | 42,686 | 11 | 0.026 | 504 |
| *C. coralliophila* A | 62,543 | 3 | 0.005 | 22,687 |
| *C. coralliophila* B | 45,264 | 46 | 0.102 | 3,623 |
| *C. coralliophila* C | 58,846 | 17 | 0.029 | 1,252 |
| *R. odorabile* A | 76,275 | 14 | 0.018 | 2,600 |
| *R. odorabile* B | 31,373 | 9 | 0.029 | 1,169 |
| *R. odorabile* C | 34,669 | 3 | 0.009 | 983 |
| *C. concentrica* A | 27,990 | 6 | 0.021 | 4,403 |
| *C. concentrica* B | 26,734 | 8 | 0.030 | 6,531 |
| *C. concentrica* C | 68,642 | 13 | 0.019 | 4,704 |
| *Scopalina* sp. A | 25,395 | 21 | 0.083 | 3,190 |
| *Scopalina* sp. B | 30,982 | 21 | 0.068 | 3,633 |
| *Scopalina* sp. C | 28,414 | 27 | 0.095 | 2,437 |
| *T. anhelans* A | 15,972 | 11 | 0.069 | 3,524 |
| *T. anhelans* B | 15,949 | 10 | 0.063 | 3,353 |
| *T. anhelans* C | 11,789 | 7 | 0.059 | 1,910 |
| *Stylissa* sp. 445 A | 32,276 | 25 | 0.077 | 1,756 |
| *Stylissa* sp. 445 B | 21,243 | 2 | 0.009 | 1,012 |
| *Stylissa* sp. 445 C | 24,377 | 0 | 0.0 | 0 |

by relative abundance counts for each functional category. Proteins with more than one assigned functional category or domain contributed to each function/functional domain equally. Profiles of the KEGG and COG pathways for each sample were calculated from the abundance of all affiliated specific functions. Abundance of functional categories belonging to more than one pathway were equally divided between the pathways.

Heatmaps, community diversity and statistical analysis were conducted in *R Core Team (2014)* using the vegan package (*Oksanen et al., 2013*).

## RESULTS

### Viral diversity associated with microbial cells of sponges

Viral sequences made up a small percentage (0–0.47%) of the total sequences within each metagenomic dataset (Table 1). In total, 451 viral sequences were identified using VirSorter's most stringent criteria (*Roux et al., 2015*), of which 344 could be taxonomically classified to species level using Metavir's LCA strategy (*Roux et al., 2014*). Cellular metagenomes from seawater contained on average 16 viral sequences per prokaryotic genome and the sponges *Scopalina* sp., *C. coralliophila*, *Stylissa* sp. 445, *C. concentrica*,

Peerj

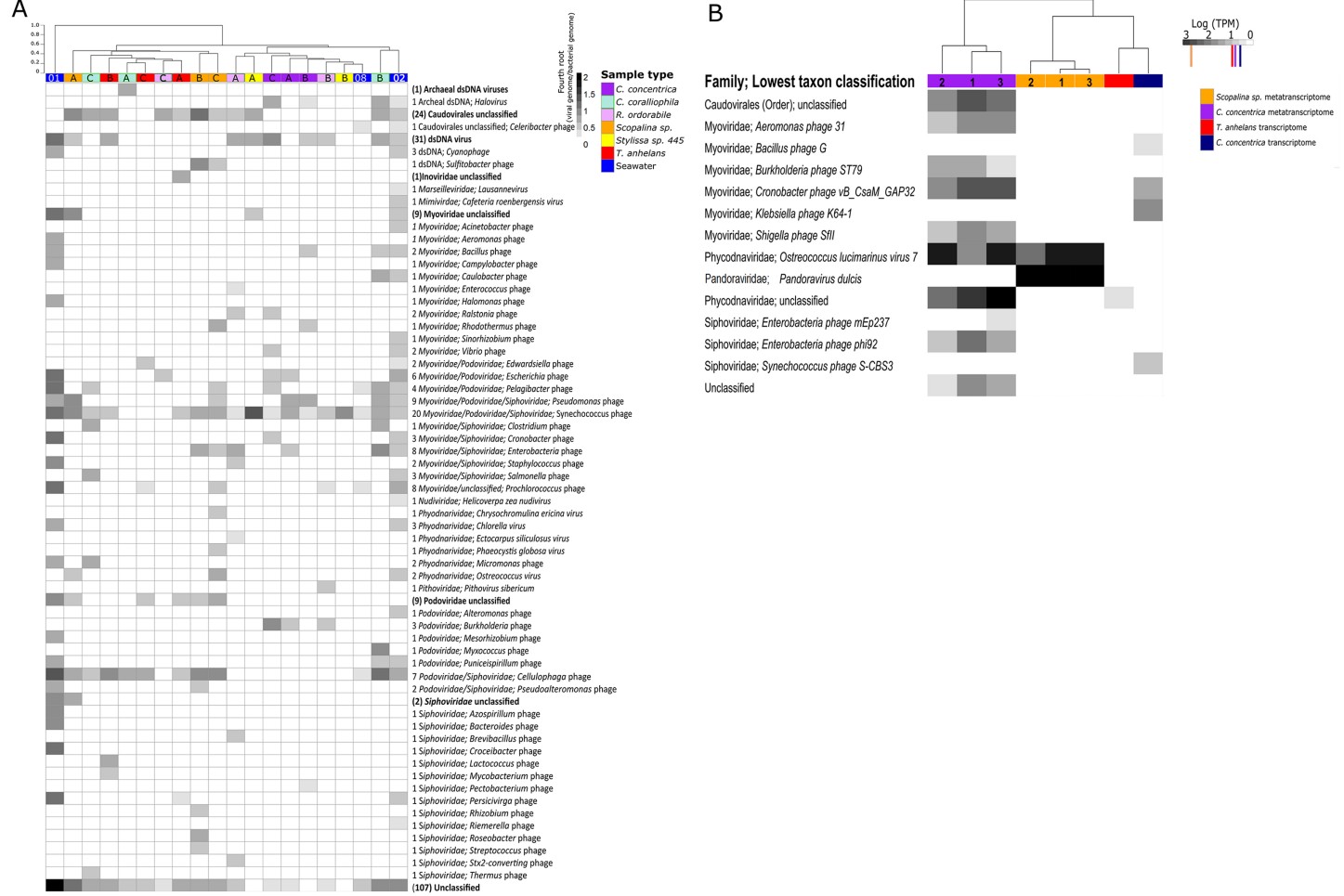

**Figure 1 Viral communities identified within sponges and seawater metagenomes (A) and (meta-) transcriptomes (B).** Samples were clustered based on Bray–Curtis dissimilarities of taxonomic profiles (tree scale indicates dissimilarity percentages). (A) Values are normalised to viral genomes per prokaryotic genome transformed with fourth root. Numbers before taxonomic assignments indicate the number of viral species found within each taxon. Taxa highlighted in bold indicate unclassified taxa above genus level and corresponding numbers in brackets indicate the number of sequences. Alphabet A, B and C indicate sample replicates for sponges and number 01, 02 and 08 indicate replicates for seawater samples, corresponding the nomenclature in *Fan et al. (2012)*. Numbers 1, 2, and 3 indicate individual sponge metatranscriptome replicates. (B) For transcriptome data, three replicates were pooled together before sequencing. Values are natural logs of normalised data transcripts per million (TPM), coloured markers on the scale represent average TPM of all transcripts in the corresponding samples coded by colours, orange: *Scopalina* sp. metatranscriptome, red: *T. anhelans* transcriptome, purple: *C. concentrica* meta-transcriptome and navy: *C. concentrica* transcriptome.

*T. anhelans* and *R. odorabile* contained an average sum of 3.6, 2.7, 2.5, 1.2, 1.1 and 0.68 contigs per prokaryotic genome, respectively.

Viromes from all samples were dominated by members of the order Caudovirales. Members of the family *Myoviridae* had the highest relative abundances in seawater followed by *Siphoviridae* and *Podoviridae*. The relative abundances of these three Caudovirales families differed among each sponge species (Fig. 1A; Fig. S2). Alpha diversity as measured by Shannon's indices on the species and family level showed seawater to contain the highest viral diversity, followed by *Scopalina* sp., *C. coralliophila*, *T. anhelans*, *R. odorabile*, *C. concentrica* and *Stylissa* sp. 445 (Table S1). High variability

**Table 2 Summary of the number of viral sequences found in the meta/transcriptome of *C. concentrica*, *Scopalina* sp. and *T. anhelans*.**

| Sample | No. of total sequences | No. of viral sequences | % Of viral sequences in metagenome | Average length of viral sequences (bp) |
|---|---|---|---|---|
| Meta-transcriptome *C. concentrica* | 383,648 | 12 | 0.003 | 2,953 |
| Meta-transcriptome *Scopalina* sp. | 94,173 | 2 | 0.002 | 3,293 |
| Transcriptome *C. concentrica* | 184,053 | 4 | 0.002 | 2,614 |
| Transcriptome *T. anhelans* | 129,529 | 1 | 0.0005 | 3,313 |

among individual replicates and between different host species was observed in viral assemblages at both the species and family levels (Fig. 1; Figs. S2–S4). Significant differences in the taxonomic structure of the viral communities across all sponge species was seen at the species and family levels (PERMANOVA, Df = 5, F.Model = 1.6254, $R^2$ = 0.42489, $P$ = 0.001 and F.Model = 1.918, $R^2$ = 0.46575, $P$ = 0.014, respectively), however no support was found for significant pairwise differences between sponge species (Table S2). Additionally, PERMANOVA did not reveal any significant differences between taxonomic structure of sponge and seawater viral assemblages at the family or species level (PERMANOVA, Df = 2; F.Model = 1.1182, $R^2$ = 0.11626, $P$ = 0.243 and F.Model = 1.4267, $R^2$ = 0.14373, $P$ = 0.111, respectively).

## Viral communities in the meta-transcriptome and transcriptome

The percentage of viral contigs in the metatranscriptomes of *C. concentrica* and *Scopalina* sp. were 0.003% and 0.002%, respectively, and no viral contigs could be identified in the metatranscriptome of *T. anhelans*. Low percentages of viral contigs were also identified in the host transcriptome of *C. concentrica* and *T. anhelans* (0.002% and 0.0005%, respectively) (Table 2) and no viral contigs could be identified in the transcriptome of *Scopalina* sp. Only DNA viruses were identified in the (meta-)transcriptomic data, with 12 viral sequences identified in the metatranscriptome of *C. concentrica* and two in *Scopalina* sp., while five viral sequences were identified in the transcriptome of *C. concentrica* and a single sequence was identified in the transcriptome of *T. anhelans* (Table 2).

The viral metatranscriptome of *C. concentrica* was primarily comprised of Caudovirales belonging to the families *Myoviridae* and *Siphoviridae*, but also contained representatives from the eukaryotic virus family *Phycodnaviridae* (Fig. 1B). In *C. concentrica*, a relative of *Ostreoccoccus lucimarinus* virus 7 (family *Phycodnaviridae*) had an average expression greater than three-fold the average total TPM of the metatranscriptome (Fig. 1B). Metatranscriptomes from *Scopalina* sp. contained two eukaryotic viruses, with closest similarity to *Pandoravirus dulcis* and *Ostreoccoccus lucimarinus* virus 7, which were expressed approximately equal to the average TPM of the metatranscriptome. (Fig. 2). Expression of *Phycodnaviridae* genes was consistent with the presence of *Phycodnaviridae* contigs in the *Scopalina* sp. metagenome (Fig. 1A).

Four sequences in the *C. concentrica* transcriptome were taxonomically assigned to bacteriophages of *Cronobacter*, *Klebsiella*, *Bacillus* and *Synechococcus*. The *Cronobacter* phage sequence in the *C. concentrica* transcriptomes was expressed at approximately twice

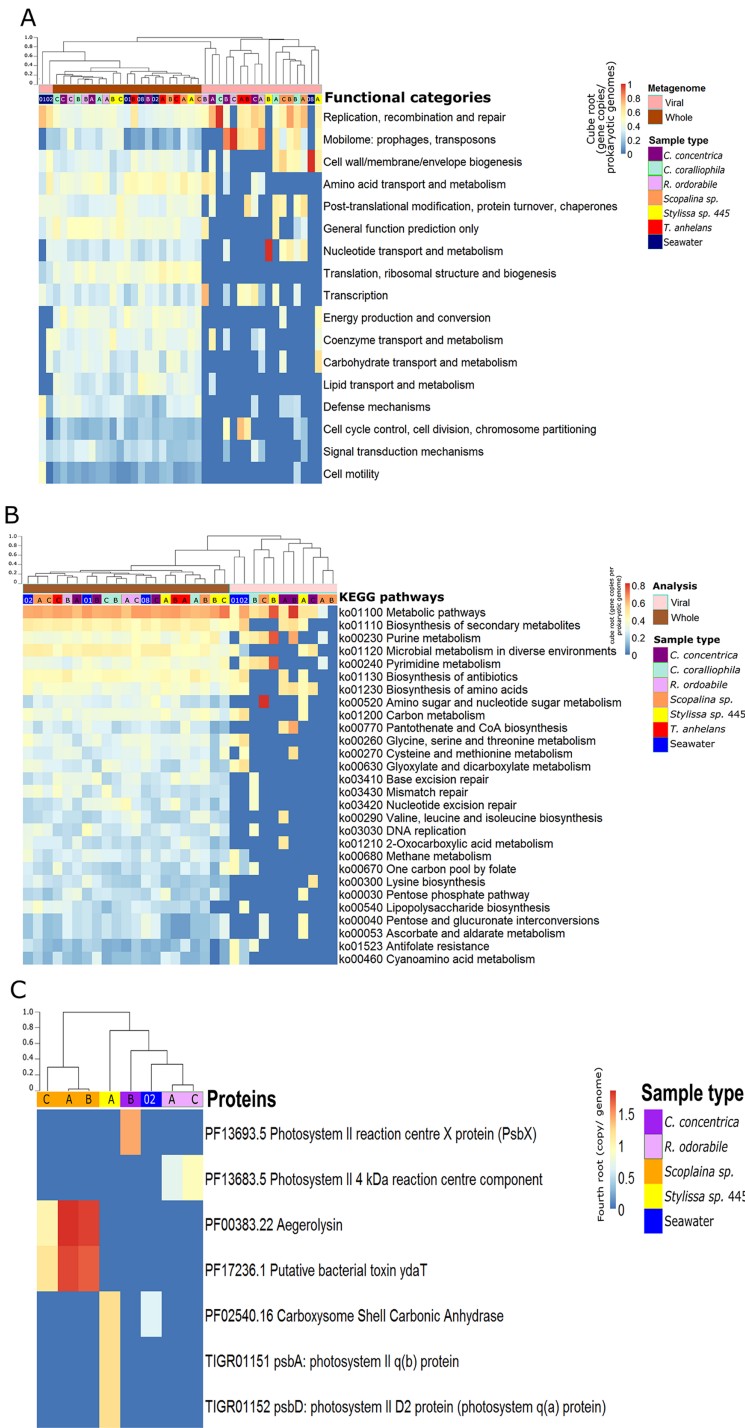

**Figure 2 Functional composition of metagenome-derived viral communities.** Functional enrichment based on COG annotations (A), KEGG pathway annotations (B) and PFAM and TIGRFAM database (C). For all analyses, annotations were assigned to assembled predicted genes normalised to the proportion of gene copies per prokaryotic genome and cube-root transformed. (A) and (B) contain annotations to the corresponding whole microbial metagenomes (viral + bacterial) for comparative purposes. Samples are clustered with Bray-Curtis dissimilarity using the 'average' method, tree scale is based on dissimilarity percentages. Letters A, B and C indicate individual sponge replicates and numbers 01, 02 and 08 indicate seawater replicates.

the average TPM, and sequences belonging to this phage taxon were also observed in the metagenomic and metatranscriptomic datasets of this sponge. Sequences assigned to *Klebsiella, Bacillus* and *Synechococcus* phages were not found in corresponding metatranscriptome or metagenome datasets. The *T. anhelans* transcriptome also contained a viral sequences assigned to the family *Phycodnaviridae* (Fig. 1B).

## Functional gene analysis of viral contigs

Functional gene annotation showed that 16%, 20%, 21.4% and 13% of all predicted genes within viral sequences could be mapped to the KEGG, COG, PFAM and TIGRFAM databases, respectively. Consistent with taxonomic analysis, functional profiles from metagenome-derived viral sequences showed high variability between individual replicates and host species (Fig. 2; Figs. S5–S8). Functional categories of the metagenomic sequences assigned to viruses were distinct from the whole (i.e. viral plus cellular) metagenomes (Figs. 2A and 2B). Not surprisingly, COG annotation associated with the categories 'mobilome: prophages, transposon' and 'replication, recombination and repair' were enriched in the viral sequences compared to the whole metagenomes (Fig. 2A). Abundant viral functions detected in the PFAM, KEGG, COG and TIGRFAM annotations comprised 'Mu-like prophage proteins,' 'DNA/RNA polymerase' and 'DNA primase and helicases' proteins (Figs. S5–S8). The KEGG pathway 'Pantothenate and CoA biosynthesis' was found in the viral metagenomes of *C. concentrica* and the 'purine and pyrimidine metabolism' pathway was found in *Stylissa* sp. 445 (Fig. 2B). Several other genes with potential auxiliary functions for bacterial or eukaryotic hosts were also detected within the viral sequences, including pathways for biosynthesis of antibiotics, genes for aegerolysin and bacterial toxin-antitoxin systems (ydaT and HicA) (Figs. 2B and 2C; Fig. S5). Genes involved in 'energy production and conversion,' and more specifically photosynthesis functions, were observed in viral sequences from the *C. concentrica, R. odorabile* and *Stylissa* sp. 445 metagenomes (Fig. 2C; Fig. S5).

Functional analysis based on (meta-)transcriptomics data showed the relative abundance of expressed genes associated with the COG category 'Post-translational modification, protein turnover, chaperones' (Fig. 3A). Viral sequences from the *C. concentrica* metatranscriptomes contained expressed genes assigned to the COG categories 'amino acid transport and metabolism' and 'transcription' and viral sequences from the *Scopalina* sp. metatranscriptomes included genes associated with the COG category 'replication, recombination and repair'. No KEGG pathways were identified for viral sequences derived from the metatranscriptomes of *Scopalina* sp (Figs. 3A and 3B). Genes associated with the KEGG pathways 'thiamine metabolism,' 'sulfur relay systems' and 'antibiotic biosynthesis' were all identified in the viral sequences in the metatranscriptomes of *C. concentrica* (Fig. 3B).

## DISCUSSION

### Viruses associated with microbial cells in sponges

Here we present the first dedicated analysis of viral sequences in a cell-separated metagenomic datasets of sponges, representing viruses attached to prokaryotic cells,

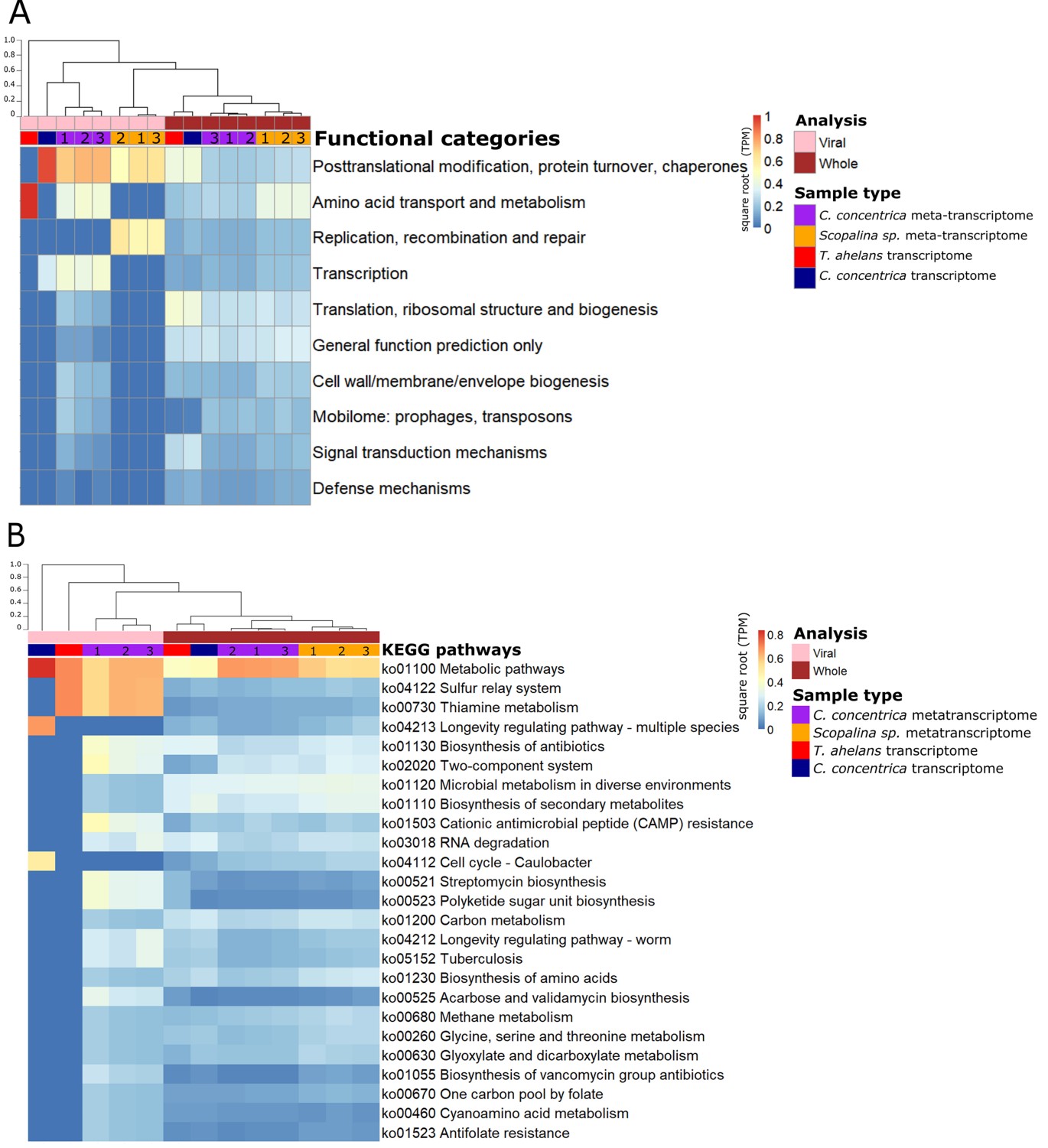

**Figure 3 Functional gene analysis of sponge viral and whole (viral and prokaryotic) meta-transcriptomes and transcriptomes.** Functional enrichment based on COG annotations (A) and KEGG pathway annotations (B) of assembled predicted genes normalised to transcripts per million (TPM) and square-root transformed. Samples are clustered with Bray-Curtis dissimilarity using the 'average' method. The tree scale is based on dissimilarity percentages. Numbers 1, 2 and 3 indicate individual sponge replicates. For transcriptomic data, the three replicates were pooled together.

virions present inside cells and/or prophages. Most bacteriophage sequences were predicted to be dsDNA viruses, although ssDNA viruses (such as *Inoviridae*) were also detected (Fig. 1A). Caudovirales were the dominant bacteriophage order, comprising the viral families *Myoviridae*, *Siphoviridae* and *Podoviridae* (*Maniloff & Ackermann, 1998*). Caudovirales have previously been reported as the dominant component of free-living metaviromes derived from sponges (*Laffy et al., 2016*, *2018*; *Weynberg et al., 2017*; *Williamson et al., 2012*) as well as viromes derived from seawater and soil (*Clokie et al., 2011*; *Zablocki et al., 2014*).

Across the viral sequences retrieved in this study, numerous matches to the Megavirales were also identified, including matches to the *Marseilleviridae*, *Mimiviridae* and *Pithoviridae* (Fig. 1A). Previous analyses of sponges also detected the ubiquitous presence of different families of Megavirales, with the authors suggesting that these viruses may infect amoeba-like phagocytic sponge cells (known as amoebocytes) or some other components of the sponge holobiont (*Laffy et al., 2019*, *2016*, *2018*). The presence of sequences assigned to Megavirales is also consistent with microscopic observations, which show 'parasitic infections' of sponge amoebocytes that resembled micrographs of *Acanthamoeba* infected with *Mimivirus* (*Claverie et al., 2009*).

Viral sequences associated with the microbial cell fraction of the sponges investigated here showed high variability within and across sponge species, both in terms of community structure and composition. This contrasts other recent studies of sponge-associated viromes performed by *Laffy et al. (2018*, *2019)*, which showed high intra-species similarity in viral assemblages. Possible explanations for this discrepancy are that *Laffy et al. (2018*, *2019)* investigated amplified DNA extracted from the free viral fraction, while our study focused on unamplified DNA from the microbial cell fraction, or that samples in the two studies were taken at different times and locations, or that the scarcity of viral DNA in the microbial cell metagenomes (*Bzhalava et al., 2018*) limited sequence coverage, which can lead to variable assembly success of viral contigs in each sample (*Smits et al., 2015*). Despite this discrepancy, detection of specific viral sequences within microbial metagenomic and (meta-) transcriptomic datasets indicate that the viral taxa and functions identified here are playing core roles within the sponge holobionts studied here, as they likely constitute dominant or active viruses.

## Active viral consortia in sponges

Active viruses in *C. concentrica* and *Scopalina* sp. were generally observed at high expression levels, equal to or above the average TPM of any gene transcript in the sample. For example, *Phycodnaviridae* and *Pandoraviridae* were observed in both sponges' (meta-) transcriptomes at 2–3 fold higher TPM values than the average TPM of other genes in *C. concentrica* and at comparable values to the average TPM in *Scopalina* sp.

*Phycodnaviridae* belong to the superfamily of nucleocytoplasmic large DNA viruses (NCLDV), which infect marine and freshwater algae (*Koonin & Yutin, 2010*) and are also prevalent in the free-virus fraction of the marine sponges *Amphimedon queenslandica* and *Ianthella basta* (*Laffy et al., 2018*). *C. concentrica* has previously been reported to contain a high abundance of diatoms (*Taylor et al., 2004*), and these diatoms could

potentially play host to these *Phycodnaviridae*, although so far no diatom-infecting NCLDV has been described in the literature. Sponges may also filter and concentrate phototrophic organisms that are host to *Phycodnaviridae* from the seawater (*Laffy et al., 2018*). The observation that the phycodnaviruses are closely related to *Ostreoccoccus lucimarinus* virus (Fig. 1B) suggests that *C. concentrica* may contain a microalgal family similar to *O. lucimarinus*, which has been found in coastal and mesotrophic systems in the Atlantic and Pacific oceans and the Mediterranean Sea (*Derelle et al., 2015*).

Expressed bacteriophage sequences were only detected in the *C. concentrica* transcriptomes and metatranscriptomes (Fig. 1B) and these were taxonomically assigned to *Cronobacter*, *Burkholderia*, *Aeromonas*, *Enterobacteria* and *Shigella* phages. Some of these bacteriophage taxa were also found in the metagenomic data (Fig. 1A). While sequences related to these bacteriophages are clearly expressed and hence suggest 'active' viruses, sequences corresponding to their bacterial hosts have so far not been found in *C. concentrica*, despite extensive metagenomic and 16S rRNA amplicon based analyses (*Esteves et al., 2016*; *Fan et al., 2012*). A potential explanation for this observation is that these viral taxa have a broader or different host-range than their taxonomic or phylogenetic affiliation implies. However this requires further investigation.

## Functional contribution of the viromes to the sponge holobiont

Approximately 80% of ORFs annotated from viral sequences could not be assigned as being homologous to proteins with known functions, which is consistent with a high percentage of 'unknown' proteins (generally >70%) commonly observed in viral metagenome studies (*Adriaenssens et al., 2015*; *Hurwitz & Sullivan, 2013*; *Laffy et al., 2018*; *Weynberg et al., 2017*; *Williamson et al., 2012*; *Zablocki et al., 2014*). Generally, genes associated with 'replication, recombination and repair', 'protein metabolism' and 'carbohydrate metabolism' were abundant in all viral metagenomes and (meta-) transcriptomes, consistent with enrichment of these functional categories in virome studies from different environments, including sponges (*Laffy et al., 2018*), corals (*Weynberg et al., 2017*), seawater (*Williamson et al., 2012*) and soil (*Adriaenssens et al., 2015*; *Zablocki et al., 2014*).

While the broad functional potential of viral communities in sponges has been previously reported, our analysis of metagenomic, transcriptomic and metatranscriptomic datasets provided a unique opportunity to identify dominant and actively expressed viral functions. Analysis of viral metagenomic sequences revealed several AMGs with potential benefits to bacterial or eukaryotic hosts, such as genes associated with the biosynthesis of antibiotics, aegerolysin, bacterial toxin-antitoxin systems (ydaT and HicA) and photosystem II (PSII) (Figs. 2A–2C). Analysis of the viral transcripts from the (meta-) transcriptomic data further revealed the expression of genes associated with antibiotic biosynthesis, photosynthesis and thiamine metabolism (Fig. 3).

A range of genes assigned to KEGG pathways for antibiotic synthesis and resistance were expressed in viruses of *C. concentrica*, including streptomycin and vancomycin biosynthesis, acarbose and validamycin biosynthesis, general biosynthesis of antibiotics and cationic antimicrobial peptide (CAMP) resistance (Fig. 3). Viruses encoding and
expressing these pathways may enhance their bacterial host's fitness through inhibition of competitor or defence against the many antibiotics that have been reported to be produced by sponge (*Faulkner, 1978*; *Hentschel et al., 2001*; *Kelman et al., 2009*; *Kim et al., 2006*; *Laport, Santos & Muricy, 2009*; *Sipkema et al., 2005*; *Torres et al., 2002*). Similarly, *Laffy et al. (2018)* reported the enrichment of herbicide-resistance genes, which were assigned to *Synechococcus* phages, in the sponge *Xestospongia testudinaria*. Herbicides are highly effective in controlling cyanobacterial populations and it was postulated that the *Synechococcus* phages confer herbicide resistance to their cyanobacterial hosts and enhanced their host's survival (*Laffy et al., 2018*).

A high relative abundance of aegerolysin genes was detected in viruses found in the *Scopalina* sp. metagenomes (average abundance of 7.8 copies per prokaryotic genome; Fig. 2C). Proteins of the aegerolysin family are widely distributed in fungi and bacteria, and to a lesser extent in plants, protozoa and insects (*Berne, Lah & Sepčić, 2009*; *Butala et al., 2017*). The functions of aegerolysins are not well understood, but they are thought to have broad-range biological properties, including antitumor, antiproliferation and antibacterial activities (*Berne, Lah & Sepčić, 2009*; *Butala et al., 2017*). Interestingly, the ascovirus *Trichoplusia ni* 2c, an obligate viral pathogen of *Pseudoplusia includens* larvae and other insects from the family *Noctuidae*, was reported to encode a hypothetical aegerolysin-like protein (*Wang et al., 2006*), suggesting that aegerolysin genes may provide benefits to the host, such as an antibacterial activity. Chemical extracts from sponges and their associated microorganisms are reported to contain antitumor, antiproliferation and antibacterial activities (*Laport, Santos & Muricy, 2009*; *Lee, Lee & Lee, 2001*; *Monks et al., 2002*; *Thakur et al., 2003*; *Yung et al., 2011*), and it may be therefore possible that viruses contribute to these features via horizontal gene transfer.

A putative bacterial toxin gene *ydaT* was found in the virome of *Scopalina* sp. at an average of 6.7 copies per prokaryotic genome (Fig. 2C) and a HicA toxin was actively expressed in the viral metatranscriptome of *C. concentrica* (Fig. S10). Toxin-antitoxin systems are widely distributed in bacteria and are associated with the formation of antibiotic-tolerant (persister) cells (*Butt et al., 2014*). YdaT belongs to the type II toxin-antitoxin systems (*Sevin & Barloy-Hubler, 2007*), where *ydaS* expresses a toxin and YdaT constitutes the antitoxin (*Yamaguchi & Inouye, 2011*). It was recently reported that YdaS and YdaT may both act as toxins by inhibiting the regular cell division of *E. coli*, which may provide resistance against β-lactamase antibiotics (*Bindal et al., 2017*). Interestingly, the HicA toxins from *Burkholderia pseudomallei* have also been reported to arrest cell growth and increase the number of persister cells tolerant to ciprofloxacin and ceftazidime (*Butt et al., 2014*). Many sponges contain high amounts of antibacterial compounds (see above) and the HicA/YdaT-type proteins may perhaps therefore play a role in providing resistance against them.

Genes for the PSII -associated proteins PsbA and PsbD were expressed in a viral contigs from *Stylissa* sp. 445 (Fig. S6), which were taxonomically assigned to *Synechococcus* phages (*Fan et al., 2012*). In addition, genes for the PS II reaction centre were identified from viral sequences identified in the *C. concentrica* and *R. odorabile* metagenomes (Fig. 2C). Interestingly, *psbA* and *psbD* have also been recently found in coral viromes (*Laffy et al., 2018*;

*Weynberg et al., 2017*) and in cyanophages infecting free-living *Synechococcus* and *Prochlorococc*us (*Lindell et al., 2004*; *Mann et al., 2003*; *Weigele et al., 2007*). Viral PSII genes are thought to supplement photosynthesis of their host and boost energy production, presumably for virion production (*Clokie et al., 2006*; *Lindell et al., 2005*). Genes encoding the carboxysome-shell carbonic anhydrase were also detected in the virome of *Stylissa* sp. 445 (Fig. 2C). Carbonic anhydrase is responsible for catalysing the conversion of carbonic acid to $CO_2$ and is utilised by cyanobacteria and marine phytoplankton to support photosynthetic carbon fixation (*Badger et al., 2005*; *Reinfelder, 2010*).

Thiamine (vitamin B1) functions as a cofactor for many essential metabolic pathways and is only produced by bacteria, fungi, and plants (*Jurgenson, Begley & Ealick, 2009*). Genes associated with thiamine metabolism were expressed in the viral metatranscriptome of *C. concentrica* (Table S3). Specifically, the expression of genes for the tyrosine decarboxylase and many cysteine desulfurase enzymes were detected (Table S3). Cysteine desulfurase is involved in the initial stages of sulfur trafficking, contributing to the addition of sulfur into the thiazole ring of thiamine (*Begley et al., 1999*; *Mihara & Esaki, 2002*) (Fig. S12). To our knowledge, this is the first time that viral-associated genes involved in thiamine metabolism have been observed, and this could potentially confer growth advantages to the host cells that cannot produce vitamin B1 (such as sponge cells). Genes associated with other vitamin synthesis pathways, such as cobalamin (vitamin B12), have also been reported in the virome of other sponges (*Laffy et al., 2018*) and in myoviruses that infect *Prochlorococcus* (*Sullivan et al., 2005*).

## CONCLUSION

Recent targeted virome analyses have discovered that marine viruses can encode AMGs previously thought to be restricted to host genomes (*Enav, Mandel-Gutfreund & Béjà, 2014*; *Hurwitz, Hallam & Sullivan, 2013*; *Laffy et al., 2018*; *Weynberg et al., 2017*). Here we demonstrate that viral AMGs can also be identified through analysis of sponge-associated metagenomic, transcriptomic and metatranscriptomic datasets, providing a unique perspective on abundant and active viruses within the sponge holobiont. Viruses contain and express AMGs for photosynthesis and vitamin synthesis, as well as for the production of and defence against toxins and antimicrobials. These viral AMGs therefore could not only contribute to the metabolic capacities of their hosts, but also potentially enhance survival of infected cells. This model suggests that viruses may play a key role in regulating the abundance and activities of members of the sponge holobiont. However further experimental work will be required to confirm or define the role that these AMGs have in the physiology and ecology of their host.

### Funding

This work was supported by funds provided by the Betty and Gordon Moore Foundation and the Australian Research Council to Torsten Thomas and an Australian Research Council Future Fellowship (FT120100480) to Nicole Webster. The funders had no role in

study design, data collection and analysis, decision to publish, or preparation of the manuscript.

## Grant Disclosures
The following grant information was disclosed by the authors:
Betty and Gordon Moore Foundation.
Australian Research Council.
Australian Research Council: FT120100480.

## Competing Interests
Torsten Thomas is an Academic Editor for PeerJ.

## Author Contributions
- Mary Nguyen conceived and designed the experiments, performed the experiments, analyzed the data, prepared figures and/or tables, authored or reviewed drafts of the paper, and approved the final draft.
- Bernd Wemheuer analyzed the data, authored or reviewed drafts of the paper, and approved the final draft.
- Patrick W. Laffy analyzed the data, authored or reviewed drafts of the paper, and approved the final draft.
- Nicole S. Webster analyzed the data, authored or reviewed drafts of the paper, and approved the final draft.
- Torsten Thomas conceived and designed the experiments, analyzed the data, authored or reviewed drafts of the paper, and approved the final draft.

## Data Availability
The raw data used here are from the previous, published studies, *Fan et al. (2012)* and *Diez-Vives et al. (2017)*.

## Supplemental Information
Supplemental information for this article can be found online at http://dx.doi.org/10.7717/peerj.10715#supplemental-information.

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
