# Peer review of "Taxonomic, functional and expression analysis of viral communities associated with marine sponges"

_PeerJ, doi:10.7717/peerj.10715_

## Round 0.1 · original submission · Major Revisions

Thank you for submitting this manuscript. We are pleased to have acquired three rapid reviews all from experts in viral metatranscriptomics, all of whom agreed that your manuscript was of high quality. Each of the reviewers provided a number of points which we request you address in a revised draft and detailed rebuttal letter outlining exactly how you have changed the manuscript to meet their suggestions.

Reviewer 1 ·

Basic reporting

Taxonomic, functional and expression analysis of viral communities associated with marine sponges
Nguyen and collaborators did an interesting work, where they studied the viral communities associated to the microbial cells of six marine sponges’ species from Australia. They used published data of metagenomes and metatranscriptomes of unfractionated viral pools (microbial cells plus associated viruses) of sponges and the surrounding seawater, and used bioinformatics tools to obtain viral contigs from these dataset.
A strong point of this article is that they analyzed the viruses associated to the microbial cells (attached to prokaryotic cells, virions present inside cells, prophages) of the sponge holobiont, and not the free viruses in the sponge. With this approach, they identified AMGs related to specific interactions between viruses and the microbial cells in the sponges. A weakness is that a small number of viral contigs (N: 451) could be recovered with confidence from the metagenomes, then limiting an overall extrapolation of the article´s findings.
Anyways, the article addresses the knowledge gap of sponges’ associated viromes using metagenomic and transcriptomic approaches, to analyze the taxonomic and functional profile of these communities. The methodology employed here, to retrieve “active” viruses associated with microbial cells was interesting and contributes with meaningful insights to the literature of sponge holobionts. Authors could have explored more in the discussion what functions are enriched in sponge viromes compared to seawater viromes, if there were any features that were significantly enriched in one of those. Maybe employing the same statistics used to analyze taxonomic structure.

The article is well written and the methods are adequate to the answer the research questions. Although, no raw data was supplied or indicated in the manuscript, and the figures need a better resolution.

Some recommendations to improve:
l. 52 It would be appropriate to add an explanation of how AMGs work here, in the introduction.
l. 76 I recommend to add a flowchart summarizing the methods (as supplementary material). As the present work uses data from published articles, with different methodologies, a flowchart can help the reader to understand how these data were generated, which data was used, and how these data were analyzed in this present article.
l. 79 and 81 When these sponges were collected (year and season)? Please, add this information.
In lines 109-114, you classify contigs into categories and choose only contigs that met categories 1 and 2 for further analysis. Maybe you could’ve kept contigs from category 3 and showed their taxonomic classification and functional properties as well, specifying that these contigs are “possible” viral predictions, since your stringent criteria only kept a few contigs for further analysis.

l. 114 Are the identified viral contigs deposited in the VirSorter database or in the NCBI? Please, add the link or NCBI id to these data.
In lines 157-165 you argue that high variability among replicates and different hosts was observed, and no significant differences between viral assemblages of sponges and seawater was found. You could’ve explored more this observation. Maybe this high variability is due to HMA/LMA dichotomy. LMA pumping rates may be higher than HMA sponges (Weisz, 2007), therefore their viromes may be somewhat similar to seawater. You could at least indicate if the sponge species in your work have been previously classified as HMA or LMA, since this is a relevant feature of sponge microbiomes and have important ecological implications.

l. 239 Another explanation can be ecological (temporal and spatial) differences between sponges, as they were sampled from different sites. I recommend the authors to explore this possibility.
l. 246 I recommend to add a consideration that the findings here (viral roles in the sponge microbiome) are valid and relevant for the studied sponges, but the extrapolation to other sponges and environments should be made with prudence, because the small number of recovered viral contigs.
l. 273 Another explanation can be that these bacteriophages are not specific to these bacterial genera, and may be infecting others similar bacteria in the sponge.
l. 355 The conclusion is well written and readily summarizes the main findings of the work. It would be interesting if the authors could add a graphical figure (model) summarizing how the viruses can affect the sponge and its microbiome, through the AMGs.
l. 365 The figures lack resolution, and the texts in the figures are difficult to read, especially in the Figure 1. I recommend enhancing the resolution of the figures.

Typing errors:
l. 261 “Phycodnaviruses” the capital letter is not needed here.
l. 298 “et al. 2002) .” remove the extra space before the dot.
l. 325 “(see above}” put this sentence between parentheses.

Experimental design

OK

Validity of the findings

OK

Additional comments

OK

Reviewer 2 ·

Basic reporting

Generally clearly reported and well-written. Small comments below.

Line 119: Awkward “and, when present, being single-copy in ≥95% of all genomes”

Line 247: Change “is indicative” to “indicate”

Lines 260-261: Awkward “The taxonomic identification made here…being closely related to”

Lines 265-268: break into two sentences

Line 273: Fragment “sponge, however this requires further investigation”

Line 356: Add comma before “as well as”

Line 358: Clarify “this”

Fig. 1 legent: Add “(B)” before describing Fig. 1B.

Fig. 1A: Text for taxonomic annotations might be too small; legend is missing for top dendrogram and heat map shading.

Fig. 2A & B: Consider adding legend to heat map shading. Are the rows ordered by fold difference between viral and whole assemblages? If not I would consider reordering by the degree of functional enrichment in viral assemblages.

Fig. 2C: On the heat map legend, typo “fouth root”

Fig. 3A & B: Legend is missing for the top dendrogram and heat map shading.

Experimental design

Research fits well within the scope of the journal. Metagenomic and metatranscriptomic workflows for viral genome curation, transcript mapping, and annotation look reasonable.

Validity of the findings

Lines 203-208, 286-291, 316-326, 339-349, and possibly others: major sections of the discussion interpret putative AMGs observed in viral sequences. How reliable are KEGG pathways in determining gene functions? Previous work suggests false positives in inferring gene function from viral metagenomic annotations, particularly for AMGs involved in antibiotic resistance. See doi:10.1038/ismej.2016.90. It’s unclear to me if the domain-specific annotations discussed in this referenced paper differ from KEGG pathways used for functional annotations in this manuscript. If KEGG annotations were more reliable, I would add this justification here or in the methods. Otherwise, I would reword this section towards a more cautious interpretation.

Lines 243-245: Some clarification needed. It’s unclear to me here why cell-associated viruses would be more variable through kill-the-winner dynamics, if these dynamics assume that cell-associated viruses are concurrently producing free-virions. My initial interpretation of the text is - do you mean that free viruses that are “stably embedded” are simply filtered by the sponge, are not infecting any cellular hosts inside the sponge, and are therefore stable through inactivity? If so, this idea seems inconsistent with lines 246-248.

Lines 271-273: “One potential explanation for this observation is that the ‘active’ bacterial viruses and their bacterial hosts may be derived from mammalian/terrestrial contaminants that are accidentally filtered into the sponge” – it’s unclear to me how this explanation is relevant to finding active viral transcription in a sample without detectable host sequences. Regardless of origin, how can a virus be transcriptionally active without its host?

Lines 289-291: “Analysis of the viral transcripts from the (meta-) transcriptomic data further revealed the expression of genes associated with antibiotic biosynthesis, photosynthesis and thiamine metabolism (Figure 3)” – since virus-encoded AMGs are derived from their host, how do we know that these transcripts are viral? Are there protein alignment trees showing that viral AMGs partition into their own “viral” clade that are distinct from their hosts? Please clarify.

Lines 303-305: “A high relative abundance of aegerolysin genes was detected in viruses found in the Scopalina sp. metagenomes (average abundance of 7.8 copies per prokaryotic genome; Figure 2C). Proteins of the aegerolysin family are widely distributed in fungi and bacteria…” – similar to the comment above, how do we know that these 7.8 copies are viral? Are they co-located on contigs with viral marker proteins? Broad in silico identification programs, such as VIRSorter, could pick up other mobile genetic elements that have distinct genome structures from that of prokaryotes, generating false positives in viral contig ID. Additional stringency measures, such as only counting AMGs if they are co-located on a contig with a known viral marker protein, would support these quantitative statements about viral AMG copy numbers per genome.

Additional comments

This manuscript exploring cell-associated viruses in sponge microbiomes is generally clear, well-written, and well-justified. A couple of sections in the discussion requires clarification - more details provided in the “validity of findings” section.

·

Basic reporting

No comment.

Experimental design

No comment.

Validity of the findings

I believe the interpretation of the findings are sound - however, see my detailed comments on several minor points.

Additional comments

PeerJ review:
In this manuscript the authors have performed phylogenetic and functional characterization of sponge associated viruses through analysis of metagenomics and metatranscriptomic datasets. The manuscript is well-written with sufficient details on methods and the results. I don’t have any major suggestions – only a few minor details that the authors can take care of.

Minor comments:
Line 93: rRNA should be RNA
Line 97: I understand that the authors tried to designate different terms for the two fractions, but simply using ‘transcriptomic’ might be a little confusing – since metatranscriptomic is a general term for environmental RNA-seq. For example, what does ‘metatranscriptomic’ mean at line 187? Is it only the rRNA depleted fraction?
Perhaps they could assign ‘polyA metatranscripotmic’ and ‘non-polyA metatranscriptomic’ for these two fractions?
Line 177: Please rephrase this statement to clarify that the virus is a ‘close relative to O. lucimarinus virus 7’, not exactly the same virus.
Line 182 – 183: Any indication that the exact same viral contigs found in metagenome were expressed? It might be interesting to match the similarity of the metagenomics and metatranscriptomic viral contigs to investigate this.
Line 258-259: I would suggest that the authors specifically mention that no NCLDV infecting diatoms is known yet. However, I understand that the possibility cannot be completely ruled out.
General comment regarding the “Functional contribution of the viromes to the sponge holobiont” section:
While I appreciate the detailed discussion on the functional implications of the viral genes – an important point is, many of these genes might have a completely different function relevant to the viral fitness and in the context of host-virus interactions. The discussion if fine as it is, but the authors might want to add a short paragraph discussing the caveats of interpreting the functions of AMGs in the viruses – most of these have not been confirmed in lab yet, and these genes might perform a different function in the virus than the known functions.

---

## Round 0.2 · Minor Revisions

Two of the three reviewers felt that the manuscript was ready for publication, but please document edits you make to a revised version in response to the suggestions (with which I concur) of the third reviewer.

Reviewer 1 ·

Basic reporting

Authors have implemented all suggested modifications.

Experimental design

corrected accordingly.

Validity of the findings

corrected accordingly.

Additional comments

congratulations on the relevant study.

Reviewer 2 ·

Basic reporting

Original comment and response in double quotations, and my new comment below.

““
Comment:
Line 119: Awkward “and, when present, being single-copy in ≥95% of all genomes”

Response:
We have rephrased this as follows:”… and being found as single-copies in ≥95% of all genomes.”
””

Change being to were?

Experimental design

ok

Validity of the findings

The following two responses did not fully address my initial discussion points. Original comments and responses are in double quotations, and my new comments are below.

““
Comment:
Lines 289-291: “Analysis of the viral transcripts from the (meta-) transcriptomic data further revealed the expression of genes associated with antibiotic biosynthesis, photosynthesis and thiamine metabolism (Figure 3)” – since virus-encoded AMGs are derived from their host, how do we know that these transcripts are viral? Are there protein alignment trees showing that viral AMGs partition into their own “viral” clade that are distinct from their hosts? Please clarify.

Response:
All viral transcripts presented here belonged to category 1 (“most confident” viral predictions) or 2 (“likely” viral predictions) by meeting Virsorter’s criteria 1 (presence of viral hallmark genes) and/or 2 (enrichment in virus-like genes) (see Methods). The identified expressed AMGs were therefore part of a polycistronic viral transcript.
””

VIRSorter would identify longer viral contigs, not individual short transcripts. It is unclear to me how one can determine that transcripts (short mRNA reads) themselves are viral unless one is certain that the gene (in this case AMG) that the transcript is mapping to is distinct from other non-viral sequences. My concern is that, considering that AMGs are derived from viral hosts, it is possible that host and viral AMGs are similar enough that host transcripts (short reads) are mapping to the viral AMG, generating false positives. Going back to my original question, have you looked at gene alignments indicating that viral AMGs are distinct from that of their hosts (at least below the mapping identity threshold)? If not, I am not convinced that one can conclude that these AMG transcripts are viral.

““
Comment:
Lines 303-305: “A high relative abundance of aegerolysin genes was detected in viruses found in the Scopalina sp. metagenomes (average abundance of 7.8 copies per prokaryotic genome; Figure 2C). Proteins of the aegerolysin family are widely distributed in fungi and bacteria…” – similar to the comment above, how do we know that these 7.8 copies are viral? Are they co-located on contigs with viral marker proteins? Broad in silico identification programs, such as VIRSorter, could pick up other mobile genetic elements that have distinct genome structures from that of prokaryotes, generating false positives in viral contig ID. Additional stringency measures, such as only counting AMGs if they are co-located on a contig with a known viral marker protein, would support these quantitative statements about viral AMG copy numbers per genome.

Response:
Please see our response to the previous comments. Yes, AMGs would have been on the same transcript as known viral marker genes.
””

A single transcript (transcriptomic short read) cannot contain a full-length AMG and known viral marker gene. Similar to the comment above, I’m not sure that one can make quantitative assessments about viral gene abundance ratio (copies per genome) without information that the gene is specifically viral, and not recruiting host reads.

·

Basic reporting

The authors have taken care of all the concerns that I raised. I believe the manuscript is suitable for publication.

Thank you.

Experimental design

I originally did not have any concern regarding the experimental design.

Validity of the findings

The authors have incorporated the improvements that I suggested in the results and conclusions.

---

## Round 0.3 · accepted · Accept

Thanks for addressing the final recommendations. Best wishes.

Reviewer 2 ·

Basic reporting

no comment

Experimental design

no comment

Validity of the findings

Thank you for clarifying the misunderstanding.

Additional comments

I support this paper for publication. Best of luck!